# Salvage Surgical Resection after Linac-Based Stereotactic Radiosurgery for Newly Diagnosed Brain Metastasis

**Ryosuke Matsuda [1,*], Takayuki Morimoto [1], Tetsuro Tamamoto [2,3], Nobuyoshi Inooka [2], Tomoko Ochi [4], Toshiteru Miyasaka [4], Shigeto Hontsu [5], Kaori Yamaki [2], Sachiko Miura [2], Yasuhiro Takeshima [1], Kentaro Tamura [1], Shuichi Yamada [1], Fumihiko Nishimura [1], Ichiro Nakagawa [1], Yasushi Motoyama [1], Young-Soo Park [1], Masatoshi Hasegawa [2] and Hiroyuki Nakase [1]**

[1] Department of Neurosurgery, Nara Medical University, Kashihara 634-8521, Japan; t.morimoto@naramed-u.ac.jp (T.M.); takeshim@naramed-u.ac.jp (Y.T.); ktamura@naramed-u.ac.jp (K.T.); syamada@naramed-u.ac.jp (S.Y.); fnishi@naramed-u.ac.jp (F.N.); nakagawa@naramed-u.ac.jp (I.N.); myasushi@oph.gr.jp (Y.M.); park-y-s@naramed-u.ac.jp (Y.-S.P.); nakasehi@naramed-u.ac.jp (H.N.)

[2] Department of Radiation Oncology, Nara Medical University, Kashihara 634-8521, Japan; ttamamo@naramed-u.ac.jp (T.T.); n-inooka@naramed-u.ac.jp (N.I.); yamaki.k@naramed-u.ac.jp (K.Y.); sachimiu@naramed-u.ac.jp (S.M.); hasegawa@naramed-u.ac.jp (M.H.)

[3] Department of Medical Informatics, Nara Medical University Hospital, Kashihara 634-8522, Japan

[4] Department of Radiology, Nara Medical University Hospital, Kashihara 634-8522, Japan; cho3co7@naramed-u.ac.jp (T.O.); tmiyasaka@naramed-u.ac.jp (T.M.)

[5] Department of Respiratory Medicine, Nara Medical University Hospital, Kashihara 634-8522, Japan; hontsu@naramed-u.ac.jp

[*] Correspondence: rmatsuda@naramed-u.ac.jp; Tel.: +81-744-22-3051

**Abstract:** Background: This study aimed to assess the clinical outcomes of salvage surgical resection (SSR) after stereotactic radiosurgery and fractionated stereotactic radiotherapy (SRS/fSRT) for newly diagnosed brain metastasis. Methods: Between November 2009 and May 2020, 318 consecutive patients with 1114 brain metastases were treated with SRS/fSRT for newly diagnosed brain metastasis at our hospital. During this study period, 21 of 318 patients (6.6%) and 21 of 1114 brain metastases (1.9%) went on to receive SSR after SRS/fSRT. Three patients underwent multiple surgical resections. Twenty-one consecutive patients underwent twenty-four SSRs. Results: The median time from initial SRS/fSRT to SSR was 14 months (range: 2–96 months). The median follow-up after SSR was 17 months (range: 2–78 months). The range of tumor volume at initial SRS/fSRT was 0.12–21.46 cm$^3$ (median: 1.02 cm$^3$). Histopathological diagnosis after SSR was recurrence in 15 cases, and radiation necrosis (RN) or cyst formation in 6 cases. The time from SRS/fSRT to SSR was shorter in the recurrence than in the RNs and cyst formation, but these differences did not reach statistical significance ($p = 0.067$). The median survival time from SSR and from initial SRS/fSRT was 17 and 74 months, respectively. The cases with recurrence had a shorter survival time from initial SRS/fSRT than those without recurrence ($p = 0.061$). Conclusions: The patients treated with SRS/fSRT for brain metastasis need long-term follow-up. SSR is a safe and effective treatment for the recurrence, RN, and cyst formation after SRS/fSRT for brain metastasis.

**Keywords:** brain metastasis; stereotactic radiosurgery; fractionated stereotactic radiotherapy; salvage surgical resection; cyst formation; recurrence; radiation necrosis

## 1. Introduction

Based on advances in chemotherapy and radiotherapy against cancer in the modern era, physicians have more options for treating brain metastases. Brain metastasis is the most common intracranial malignant tumor in adults, occurring in up to 8–35% of all cancer patients [1].

Treatment for brain metastasis includes whole brain radiotherapy, surgery, and SRS/fSRT. In particular, to manage smaller brain metastases, SRS is the first-line option. SRS/fSRT for brain metastasis are linked to good tumor control and fewer complications [2–4]. In our previous studies, SRS and fSRT, using a frameless fixation system for brain-stem metastasis and large brain metastasis with unsuitable surgical resection, showed good tumor control with the possibility of reducing radiation necrosis (RN) [5,6].

However, among the long-term survivors, thanks to the advance of chemotherapy, a few patients previously treated with SRS/fSRT have shown recurrence, RN, or cyst formation in long-term follow-up after SRS/fSRT [7–11]. The optimum treatment option for patients with recurrence in a previously irradiated field remains controversial. In some studies, re-irradiation for the recurrence of brain metastasis after SRS was reported [7,9,12–16]. In contrast, another study recommended salvage surgical resection (SSR)[8,17–19]. Less attention has been paid to re-treatment for recurrence, RN, and cyst formation than to initial SRS/fSRT for brain metastasis, because disease progression of the primary cancer may not permit long-term survival in patients treated with SRS/fSRT.

In this retrospective study, we examined 21 consecutive patients treated with SSR for brain metastasis after SRS/fSRT to assess the efficacy and limitations of SSR.

## 2. Materials and Methods

### 2.1. Patient Characteristics

Clinical data were retrospectively collected to evaluate the efficacy and limitations of SSR among patients treated with SRS/fSRT for newly diagnosed brain metastasis. The ethical committee of Nara Medical University (Kashihara, Japan) approved this retrospective study in May 2020 (No. 2634). Between November 2009 and May 2020, we treated 335 consecutive patients with 508 SRS/fSRTs for 1146 brain metastases at our hospital. Patients who needed SRS/fSRT for the recurrence of previously irradiated brain metastasis were excluded from the study. Patients who needed SRS/fSRT for resected cavity after initial surgical resection of newly diagnosed brain metastasis were also excluded. Eventually, 318 patients with 484 SRS/fSRTs for 1114 brain metastases were included in this study. Then, between February 2011 and December 2020, we treated 21 consecutive patients with SSR after SRS/fSRT for brain metastasis at our hospital. Prior to SRS/fSRT, each patient was evaluated by the tumor board review on brain tumors—a multidisciplinary team including neurosurgeons, neuro-radiologists, and radiation oncologists—to determine the most appropriate therapy. Table 1 lists all patients' clinical characteristics (Table 1).

**Table 1.** The characteristics of all patients with SSR after SRS/fSRT.

| Characteristic | Nubmer |
|---|---|
| Sex | |
| man/woman | 12/9 |
| Age at the salvage surgical resection (years) | |
| median (interquatile range) | 69 (63–71) |
| Tumor origin | |
| lung/breast/colon/thyroid | 16/3/1/1 |
| Tumor location | |
| frontal/cerebellum/occipital/temporal/parietal | 10/6/3/1/1 |
| Brain metastasis at the initial SRS/fSRT | |
| single/multiple | 9/12 |
| Maximum tumor diamter(mm) at the initial SRS/fSRT | |
| median (interquatile range) | 11 (5–18) |
| Tumor volume(cm$^3$) at the initial SRS/fSRT | |
| median (interquatile range) | 1.02 (0.21–2.31) |
| Pathology | |
| recurrence/radiation necrosis or cyst formation | 15/6 |
| Driver mutation in lung cancer | |
| EGFR/ALK/negative/NA | 2/2/7/5 |

EGFR: epidermal growth factor recptor, ALK: anaplastic lymphoma kinase, NA: not available.

### 2.2. SRS and fSRT

Planning of SRS and fSRT was based on a computed tomography (CT) scan with a slice thickness of 1 mm. All patients were immobilized in a thermoplastic mask. The gross tumor volume (GTV) for each lesion was delineated on MRI with a slice thickness of 1 mm. The planning target volume (PTV) was defined as GTV plus 1–2 mm for all dimensions. Treatment was provided within 1 week after planning the CT scan. Treatment planning was performed using BrainSCAN® or iPlan® RT (BRAINLAB AG, Munich, Germany). The irradiation dose was prescribed to confirm a dose coverage of 90% for the PTV. Dose calculations were performed using a pencil beam algorithm. SRS and fSRT were performed using linacs with a micro-multi-leaf collimator: Novalis® (BRAINLAB AG, Munich, Germany), with a collimator width of 3 mm, or TrueBeam® STx (Varian Medical Systems, Palo Alto, CA, USA), with a collimator width of 2.5 mm. Nineteen patients were treated with Novalis and two patients were treated with TrueBeam STx. Every patient was treated with X-rays of 6 MV beam energy.

Patient positioning and verification were performed using BrainLab ExacTrac® (BRAINLAB AG, Munich, Germany). This device comprises two infrared cameras and two dual diagnostic kV X-ray tubes, which can be moved automatically into treatment position to minimize setup errors [20,21].

Basically, patients with neurological symptoms, and patients with brain metastases larger than 20 mm in size, underwent fSRT. Asymptomatic patients with brain metastases smaller than 20 mm were treated with SRS, and the decision was based on tumor size, location, surrounding edema, and other reasons.

All patients were treated using Novalis and TrueBeam STx with 18–23 Gy in a single fraction for SRS or 30–42 Gy in 3–6 fractions for fSRT via non-coplanar multi-beams, non-coplanar multi-arcs, or both. The treatment methods in SRS or fSRT were conformal beams, dynamic conformal arcs, intensity-modulated radiotherapy (IMRT), or hybrid arcs. Hybrid arcs is a novel treatment technique blending aperture-enhanced optimized arcs with discrete IMRT elements, thereby allowing arc selection with a set of static IMRT beams [22].

### 2.3. Surgical Procedures

All patients treated with SRS/fSRT for brain metastasis were regularly evaluated with MRI, including perfusion-weighted images at intervals of 3 months after initial SRS/fSRT. Once disease progression was detected, MRI was performed at intervals of 1–2 months. The decision for craniotomy and resection for disease progression was based on evidence of clinical deterioration and associated imaging progression judged by the tumor board review on brain tumors. Neuroimaging indications for SSR included an enlarging lesion with increased cerebral blood flow in perfusion-weighted images of MRI, hemorrhage, and symptomatic mass effect unresponsive to medical management with corticosteroids. All SSRs except one were performed under general anesthesia. One patient underwent an awake craniotomy. Image-guided surgeries using intraoperative ultrasonography (HITACHI ALOKA, Japan) were performed in all cases. In nine cases, the BrainLab navigation system was useful to detect the tumor boundaries even though the lesions were irradiated with SRS/fSRT.

### 2.4. Statistics

The median survival time was calculated using the Kaplan–Meier method. The log-rank test was used for univariate analyses. The time from initial SRS/fSRT to SSR between the groups was compared with the Mann–Whitney U test. All of the analyses mentioned above were performed with the EZR software (Saitama Medical Center, Jichi Medical University, Saitama, Japan)[23], and $p < 0.05$ was considered to indicate statistical significance.

## 3. Results

### 3.1. Surgical Results and Pathological Diagnosis

Between November 2009 and May 2020, 318 consecutive patients with 1114 brain metastases were treated with SRS/fSRT for newly diagnosed brain metastasis at our hospital. During this study period, 21 of 318 patients (6.6%) and 21 of 1114 brain metastases (1.9%) went on to receive SSR after SRS/fSRT. The median time from initial SRS/fSRT to SSR was 14 months (range: 2–96 months). The time from SRS/fSRT to SSR was shorter in the recurrence than in the radiation necrosis and cyst formation (14 vs. 35 months), but these differences did not reach statistical significance ($p = 0.066$). Before SSR, 14 patients had some symptoms, including headache, motor weakness, speech disturbance, disorientation, and cerebellar ataxia. Among the other seven patients without any symptoms, preoperative imaging suspected recurrence in six cases and enlarged cyst formation in the cerebellum in one patient.

Permanent pathological diagnosis revealed recurrence with viable cancer cells in 15 cases (4.7%) out of 318 patients. In another 6 cases (1.9%) out of 318 patients, RN and cyst formation were diagnosed without viable cancer cells (Table 1). The median follow-up after SSR was 17 months (range: 2–78 months). One patient with recurrence died due to progression of brain metastases within 3 months; hence, follow-up MRI data at 3 months for this patient are unavailable. Among the remaining 20 patients who underwent follow-up MRI every 3 months, the examination revealed recurrence at the same site of resection in 3 cases (15%) out of 20 patients. These three patients underwent surgery again, one subsequently underwent additional whole brain radiotherapy, one underwent additional local radiotherapy, and one underwent additional fSRT for resected cavity. Among the other eleven patients with SSR for recurrence, seven patients treated with gross total resection received no further treatment, three patients underwent additional SRS/fSRT for the cavity, and one patient underwent additional whole brain radiotherapy.

Six patients treated with SSR for RN with/without cyst formation experienced no recurrence during the follow-up period. Before SSR, 14 patients had some symptoms, including headache, motor weakness, speech disturbance, disorientation, and cerebellar ataxia. Eleven patients had an improvement of their symptoms after surgery. Among the other seven patients without any symptoms, preoperative imaging suspected recurrence

in six cases and enlarged cyst formation in the cerebellum in one patient. All asymptomatic patients developed no new neurological deficits after SSR.

### 3.2. Survival Rate and Prognostic Factors

Eleven patients died at the last follow-up after SSR. Eight patients died because of worsening of extra-cranial cancer, and the remaining three patients died owing to progression of carcinomatous meningitis. The median survival times from SSR and from initial SRS/fSRT were 17 and 74 months, respectively. The median survival time from SSR in the patients with recurrence was 16 months. Those with recurrence had a shorter survival time from initial SRS/fSRT than those without recurrence, but these differences were not significant ($p = 0.061$) (Figure 1).

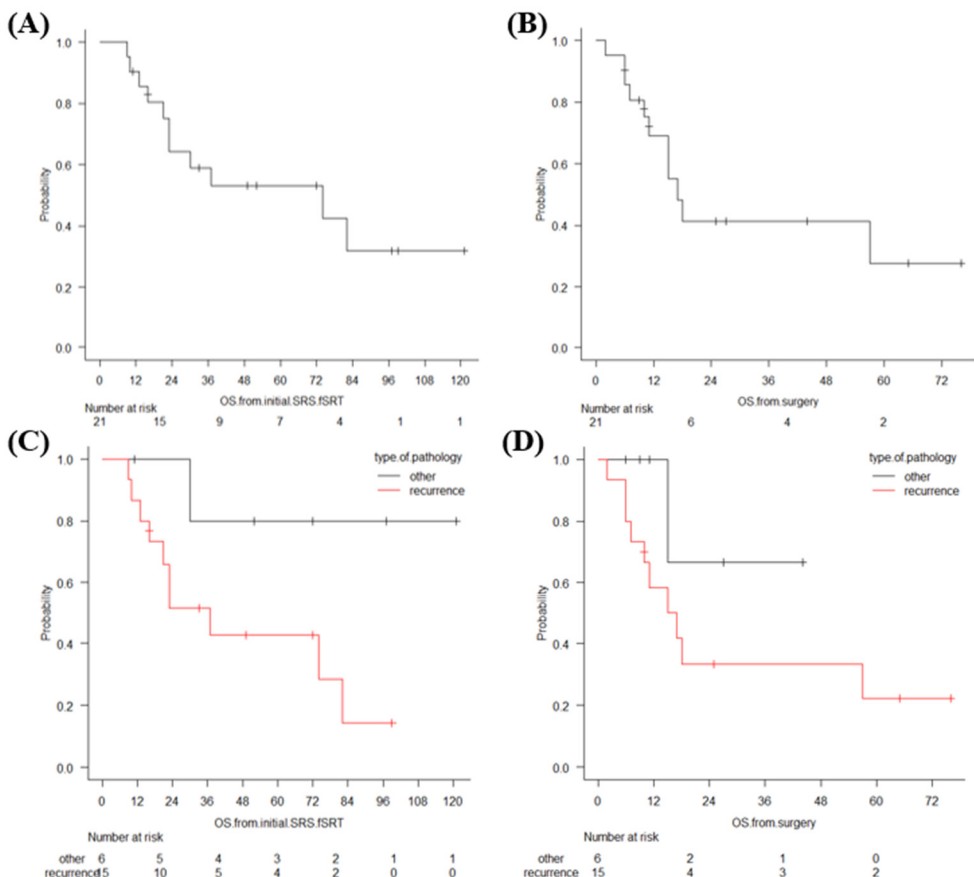

**Figure 1.** Overall survival time since initial SRS/fSRT (**A**) and overall survival since SSR (**B**), estimated using the Kaplan–Meier method. Overall survival time since initial SRS/fSRT (**C**) and since SSR (**D**) in the recurrence group and in RN/cyst formation, estimated using the Kaplan–Meier method.

### 3.3. Complications

One patient had a postoperative hemorrhage in the cavity, on day 4 after SSR for recurrence of brain metastasis from lung spindle cell carcinoma. This patient underwent evacuation of hemorrhage. No patients experienced any postoperative surgical site infection. One patient developed hydrocephalus after SSR, for which a ventriculo-peritoneal shunt was placed.

## 4. Discussion

### 4.1. Local Recurrence after SRS/fSRT for Brain Metastasis

SRS is an effective, routinely used treatment modality for brain metastasis, achieving high local tumor control (LTC) rates and typically avoiding the neurocognitive toxicities associated with whole brain radiation therapy. Based on a recent systematic review, the reported one-year LTC rates vary from 71% to >90% [3]. Nevertheless, the efficacy of SRS using a Gamma Knife (GK), in terms of LTC and complications, depends on the tumor size. In a large cohort treated with SRS, patients whose tumors at first SRS had a maximal diameter > 10 mm or a volume of 0.25 cm³ were associated with shorter overall survival [3]. Among the long-term survivors after SRS/fSRT, thanks to the advance of chemotherapy, a few patients previously treated with SRS/fSRT have shown recurrence, RN, or cyst formation in long-term follow-up after SRS/fSRT [7–11]. The optimum treatment option for patients with recurrence in a previously irradiated field remains controversial.

In this study, we reported the clinical results of SSR after SRS/fSRT for newly diagnosed brain metastasis: 21 of 318 patients (6.6%) and 21 of 1114 brain metastases (1.9%) went on to receive SSR after SRS/fSRT. Histopathological diagnosis after SSR was recurrence in 15 cases (4.7%), and RN or cyst formation in 6 cases (1.9%). The median survival time from SSR and from initial SRS/fSRT was 17 and 74 months, respectively. The cases with recurrence had a shorter survival time after initial SRS/fSRT than those without recurrence.

McKay et al. reported the recurrence of brain metastasis after GK SRS. Among 738 patients treated with GK SRS, 58 (7.85%) patients had a recurrence with local failure. Of these 58 patients, 32 underwent a second course of GK SRS [7]. Among them, 24% developed symptomatic RN and the one-year control rate was 79%. Rana et al. reported that 32 brain metastases with recurrence after linac-based SRS/fSRT were treated with linac-based salvage SRS. The median interval time between initial SRS/fSRT and second SRS was 9.7 months. The overall control rate was 84.4% with 18.8% RN [9]. Balermpas et al. reported 32 recurrent brain metastases after GK SRS and Cyber Knife SRS. The one-year local control rate was 79.5%, and the overall rate of radiological RN was 16.1% [12]. Repeated SRS for the recurrent brain metastasis after SRS or fSRT is summarized in Table 2 [7,9,12–16]. In this study, among 318 patients treated with SRS/fSRT for newly diagnosed brain metastasis, 15 (4.7%) patients were proven to be recurrent after SRS/fSRT. The proportion of recurrent cases in this study was relatively lower than in a previous report. Our data do not include cases of repeated SRS/fSRT cases after initial SRS/fSRT, as this study focuses on cases of surgical treatment after SRS/fSRT in this period.

**Table 2.** Repeated stereotactic radiosurgery for recurrent brain metastasis after stereotactic radiosurgery and fractionated stereotactic radiotherapy.

| Author (Year) | Modality | Number of pts | Repeated SRS/fSRT | Local Tumor Control Overall Survival | Radiation Necrosis |
|---|---|---|---|---|---|
| Kim (2013) [13] | TomoTherapy | 32 | TomoTherapy | 1y LTC 77% | 9.4% |
| Terakedis (2014) [14] | Linac-based SRS | 37 | Linac-based SRS | 1y LTC 80.6% OS 8.3M | 16% |
| Minniti (2016) [15] | Linac-based SRS | 43 | Linac-based fSRT 21–24Gy/ 3 fractions | 1y LTC 70% 2y LTC 60% 1y OS 37% | 19% Radiographic RN |
| Rana (2017) [9] | Linac-based SRS | 28 | Linac-based SRS | Overall LTC 88.3% | Overall RN 18.8% |
| Mckay (2017) [7] | GK | 32 | GK | 1y OS 70% 1y LTC 79% | RN 30% Symptomatic 24% |
| Balermpas (2018) [12] | GK/CK | 31 | GK/CK | 1y LTC 79.5% 1y OS 61.7% | RN 16.1% G3/4 12.9% |
| Iorio-Morin (2019) [16] | GK | 56 | GK 18Gy (12–20Gy) | 1y LTC 68% 1y OS 92% OS 14m | Radiation-induced edema 8.3% Radiation-induced necrosis 5.0% |

GK: Gamma Knife, CK: Cyber Knife, OS: overall survival, pt: patient, NA: not available, SRS/fSRT: stereotactic radiosurgery and fractionated stereotactic radiotherapy.

The results of SSR for brain metastases after SRS/fSRT, including our study, are summarized in Table 3 [8,17–19]. The median time from initial SRS/fSRT to SSR varies from 5.2 to 14 months. Surgical morbidity and mortality rates were 0–21.9% and 0–3%, respectively. The median survival time from SSR varies from 7.6 to 20.2 months. The patients who were previously treated with SRS/fSRT need careful follow-up until at least 2 years after SRS/fSRT. Compared to repeated SRS/fSRT, SSR can rapidly reduce intracranial pressure, improve neurological function, and confirm histopathological diagnosis. Once disease progression in the same site after SRS/fSRT occurred, we recommended the SSR for the accurate pathological diagnosis and rapid decompression with the lower risk of surgical morbidity and mortality.

**Table 3.** Summary of SSR for recurrent brain metastasis after SRS/fSRT.

| Author (Year) | Modality | SRS/SRT | Number of pts/Lesions | Median Time from Initial SRS/fSRT to SSR | Median Survival Time from SSR | Rate of Radiation Necrosis | Complication | Local Failure after SSR | Surgical Mortality |
|---|---|---|---|---|---|---|---|---|---|
| Vecil (2005) [17] | NA | SRS | 61/74 | 5.2 | 11.1 | 6/74(8%) | 12%(Major) 8%(Minor) | 13/74(17.6%) | 3% |
| Truong (2006) [18] | GK | SRS | 32/38 | 8.6 | 8.9 | 4/32(12.5%) | 7/32(21.9%) | 9/32(28%) | 3% |
| Kano (2009) [8] | GK | SRS | 58/58 | 7.2 | 7.6 | 0/58(0%) | 4/58(6.9%) | 18/58(31%) | 1.7% |
| Mitsuya (2020) [19] | Linac based SRS/SRT | SRS/fSRT | 48/54 | 12 | 20.2 | 7/54(13%) | 0% | 24/54(24.6%) | 0% |
| This study (2021) | Linac based SRS/SRT | SRS/fSRT | 21/24 | 14 | 17 | 4/21(19%) | 1/21(4.2%) | 4/21(19%) | 0% |

SSR: salvage surgical resection, GK: Gamma Knife, pt: patient, NA: not available, SRS: stereotactic radiosurgery, fSRT: fractionated stereotactic radiotherapy.

*4.2. Radiation Necrosis after SRS/fSRT for Brain Metastasis*

RN is an inflammatory reaction that occurs between a couple of months and several years following SRS and is one of the most common adverse effects after SRS/fSRT. In previous studies, RN after SRS/fSRT occurred in 5–25% patients [24–28]. The definition of RN varies across studies and is based on radiological findings, including perfusion-weighted MRI, MR spectroscopy, and positron emission tomography with fluorodeoxy-glucose and other tracers, as well as pathological findings after surgical resection. Therefore, it is difficult to compare the reported incidence of RN in each study.

Kohutek et al. reported that the median time from initial SRS to RN was 10.7 months (2.7–47.7 months) [24]. RN is seen as a contrast-enhancing lesion with perilesional edema at the site of previous SRS, radiologically, and can be asymptomatic or cause neurological symptoms. Commonly cited risk factors for RN include target dose and volume, previous radiotherapy, and the concurrent use of systemic agents [25].

For the management of symptomatic RN including headache, cognitive impairment, seizures, or focal deficits related to the location of RN, oral steroids are the first line of treatment [26]. Some patients need oral steroids for a long duration, but cannot continue to take steroids because of the unfavorable side effects. When RN following SRS/fSRT is resistant to oral steroids, bevacizumab—a humanized antibody inhibiting the vascular endothelial growth factor—may improve patient status and reduce the use of corticosteroids [29]. Hyperbaric oxygen therapy for radiation necrosis led to clinical and radiologic improvement or stability in patients treated with SRS/fSRT for brain metastasis [30]. For symptomatic patients with RN resistant to medication including oral steroids and bevacizumab or those with suspected recurrence, we performed SSR.

*4.3. Cyst Formation after SRS/SRT for Brain Metastasis*

Alattar et al. reported that cyst formation after linac-based SRS occurred in 0.9% of 1106 treated lesions. Among the nine patients, four who had neurologic deterioration despite steroid treatment underwent surgical fenestration and biopsy of the cyst wall [10]. Ishikawa et al. reported that the incidence of cyst formation was estimated as 10% in long-term survivors (>3 years) without tumor recurrence [31].

In the present study, there were two cases with cyst formation. The time from initial SRS/fSRT to cyst formation was 85 and 96 months, respectively. We performed fenestration of the cyst wall and removed the necrotic tissue surrounding the cyst wall. No recurrence of cyst formation occurred in these two cases. Aizawa et al. reported that cyst formation occurred 10 years after initial SRS [32]. In the long-term survivors treated with SRS/fSRT, even though follow-up MRI revealed no new brain metastasis and no recurrence, physicians should pay more attention to the development of cyst formation >10 years after SRS/fSRT. Cyst formation after SRS or SRT is summarized in Table 4 [10,31–33].

**Table 4.** Summary of cyst formation after SRS/fSRT.

| Author (Year) | Modality | Number of Patients | Time to Cyst Formation | Treatment | Prognosis |
|---|---|---|---|---|---|
| Ishikawa (2009) [31] | GK | 8 | 53 months (median) | 5 cases: Ommaya reservoir<br>2 cases: reject<br>1 case: asymptomatic | 5 cases: alive<br>2 cases: died (progressive cyst formation)<br>1 case: died with primary cancer |
| Yamamoto (2012) [33] | GK | 7 | 53 months (median) | 7 cases: Ommaya reservoir | NA |
| Aizawa (2018) [32] | Linac-based SRS | 1 | 123 months | 1 case: Ommaya reservoir | alive |
| Alattar (2018) [10] | Linac-based SRS/fSRT | 11 | 218 days | 2 cases: asymptomatic<br>3 cases: steroid<br>4 cases: surgical fenestration | alive |
| This study (2021) | Linac-based SRS/fSRT | 2 | 85 months and 96 months | 2 cases: surgical fenestration | alive |

K: Gamma Knife, SRS/fSRT: stereotactic radiosurgery and fractionated stereotactic radiotherapy, NA: not available.

The mechanism of development of cyst formation is unclear. Ishikawa et al. hypothesized that cyst formation is essentially the same as or very similar to those in patients treated with SRS for arteriovenous malformation, and thus is not the result of disease progression. Breakdown of the blood–brain barrier appears to play an important role in the cyst formation process. The relatively high blood flow volumes and increased permeability of injured blood vessel walls in the irradiated lesion may also promote cyst formation within the area of radiation-induced degeneration, continuing for several years after SRS/fSRT [31].

*4.4. Limitations*

The small sample size included in the present study and retrospective analyses does not allow us to evaluate the proper treatment for recurrence, RN, and cyst formation after SRS/fSRT. Although similar clinical studies have been recently conducted, each study involved different criteria, such as for the definition of RN and treatment modalities, hence lacking consistency in analysis. In the future, there is a need to develop a diagnostic technique that can easily differentiate between recurrence and radiation necrosis after SRS/fSRT for brain metastasis. A randomized trial to determine the conditions for treating recurrence after SRS/fSRT for newly diagnosis brain metastasis is warranted.

**5. Conclusions**

The patients treated with SRS/fSRT for newly diagnosed brain metastasis require long-term follow-up. Once disease progression in the same site after SRS/fSRT occurred, we recommended the SSR for the accurate pathological diagnosis and rapid decompression with the lower risk of surgical morbidity and mortality.



**Author Contributions:** Conceptualization, R.M., T.M. (Takayuki Morimoto), M.H. and T.T.; methodology, R.M., M.H., T.M. (Takayuki Morimoto) and T.T.; formal analysis, R.M., M.H. and T.T.; data curation, R.M., N.I., S.M., T.M. (Takayuki Morimoto), S.H. and T.T.; writing—original draft preparation, R.M., M.H., S.M. and T.T.; writing—review and editing, T.M. (Takayuki Morimoto), S.H., T.O., T.M. (Toshiteru Miyasaka), K.Y., Y.T., K.T., S.Y. F.N., I.N., Y.M., Y.-S.P. and H.N.; supervision, M.H. and H.N.; project administration, H.N. All authors have read and agreed to the published version of the manuscript.

**Funding:** This research received no external funding.

**Institutional Review Board Statement:** The Ethical Committee of Nara Medical University, Nara, Japan, approved this retrospective study in May 2020 (No. 2634).

**Informed Consent Statement:** This study is a retrospective study. Patients were not required to provide informed consent to the study because the analysis used anonymous clinical data that were obtained after each patient agreed to treatment by written consent. Additionally, we applied the opt-out method to obtain consent in this study.

**Data Availability Statement:** Data sharing is not applicable to this article as no datasets were generated or analyzed during the current study.

**Conflicts of Interest:** The authors declare no conflict of interest.

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
