# Peer review of "Salvage Surgical Resection after Linac-Based Stereotactic Radiosurgery for Newly Diagnosed Brain Metastasis"

_curroncol, doi:10.3390/curroncol28060439_

Round 1

Reviewer 1 Report

Nicely written manuscript. Minor edits are required as in the attached file.

Author Response

To Reviewer #1

In this revision, we have made some changes to the manuscript and hope that you will be satisfied with the results.

Comment 1

Thank you for your suggestion.

I adapted the sentence from “ 2 patients was” to “ 2 patients were”.

Comment 2 & 3

I adapted 1.8cm and 1.3cm instead of “mm”.

But, reviewer #2 asked us to remove two illustrative cases.

Finally, we removed two illustrative cases from our manuscript.

Comment 4

We added the sentence with hyperbaric oxygen therapy.

Reviewer 2 Report

The present study is a retrospective evaluation of salvage surgical resection following linac-based stereotactic radiosurgery. The study sample is small; however, the authors present single institutional data, collected over 10-year period. The analysis is based on 21 consecutive patients, who were treated with salvage surgical resection for 21 brain metastases, following stereotactic radiotherapy (either radiosurgery or fractionated stereotactic radiotherapy).

Firstly, I would suggest modifying the title of this manuscript, as it is including both the terms radiosurgery and radiotherapy, the title could be modified to be more concise.

Abstract

I would propose to adapt the following sentence, to make it clearer: “Histopathological diagnosis after SSR was recurrence, radiation necrosis (RN) and cyst formation in 15 and 6 cases, respectively.” (For example, “Histopathological diagnosis after SSR was recurrence in 15 cases, and radiation necrosis (RN) or cyst formation 6 cases.”

Introduction

I would like to point out a somewhat brief introduction into the researched problem. Please see also comments under the Discussion section. In general, it seems to me that the authors should place more emphasis on salvage surgery, how to select patients, how it is affected by prior imaging and what is already known in the literature.

Methods

The methods are presented in a detail and are in general appropriately described. 

Results

There is no need to repeat some of the results twice (in the text and in the Table 1).

Page 4, Rows 152-154: I would suggest reporting percentages of evaluated pathological reports / performed surgical resections (only 21 patients out of 318 underwent surgery, it would make sense to report the recurrence rate and the share of radio-necrosis from all operated metastases)

Some data from the “Materials and Methods” (2.1. Patients’ characteristics) section are repeated in the “Results” section (3.1. Surgical Results and Pathological Diagnosis)

Minor comment: 3.2. Survival rate and Prognostic Factors – I would suggest using extra-cranial cancer instead of “systemic cancer”. Please also check the spelling of “Prognostic”.

In the results section, the authors present two illustrative examples, but I do not think that such presentation is really necessary.

Discussion

In general, I would propose the authors to adapt the discussion section. For example, to start with a summary of the results (salvage surgical resection, findings, …) and then to outline the interpretation the results (SSR) in the light of the published literature. Some parts of the discussion section would fit more in the introduction section, for example section 4.1 (The explanation/the rationale of SRS/fSRT for Brain Metastasis) and partly section 4.2 (Recurrence after SRS/fSRT for Brain Metastasis) – to outline the rationale why to perform SSR, and what is already know in this area. Summary of repeated SRS for recurrent brain metastasis is of course, interesting, but I think that this section is not really necessary, as also the authors acknowledged that their data does not include cases of repeated SRS/fSRT cases after initial SRS/fSRT.

Under the section “4.3. Radiation Necrosis after SRS/fSRT for Brain Metastasis”, I am missing the discussion of the authors why the rate of radiation necrosis, reported in this series, is slightly higher than reported in the published literature.

I am also missing the authors to discuss any future directions.

Conclusions

I am not sure if the authors can conclude this article with the following sentence: “In this study, the median survival time from SSR was 17 months. SSR is a safe and effective treatment for the recurrence, RN, and cyst formation after SRS/fSRT for brain metastasis.” The conclusion sentences should be adapted.

Author Response

To Reviewer #2

In this revision, we have made some changes to the manuscript and hope that you will be satisfied with the results.

Comment1

Thank you for your suggestion. I changed the title more concisely.

Comment2

I adapted “Histopathological diagnosis after SSR was recurrence in 15 cases, and radiation necrosis (RN) or cyst formation in 6 cases.”

Comment3

We added some sentences into “Introduction” in order to improve this section.

Comment4

We removed the duplicated explanations from “2.1. Patient Characteristics”.

Comment5

We added the percentage of the recurrence rate after SSR.

Comment6

We removed the duplicated explanations from “3.1. Surgical results and Pathological diagnosis”.

Comment7

We used the “extra-cranial cancer” instead of “systemic cancer”.

We corrected the miss-spelling of “prognositic”.

Comment8

We removed two illustrative cases from this manuscript.

Comment9

Thank you for your recommendations.

We adapted the “Discussion section” as you pointed out.

Comment10

As you mentioned, in this study, the rate of radiation necrosis is slightly higher than other reports.

I think that the rate of radiation necrosis in Table3 included two cases with the cyst formation. We removed two cases with cyst formation. We adapted the rate of radiation necrosis.

Moreover, small sample size of this study did not allow to evaluate the precise occurrence of radiation necrosis. Although we removed the illustrative cases as you pointed out, but I would like to leave the Table2 (Summary of repeated SRS for recurrent brain metastasis). I think that it is very important for the readers to understand the treatment for recurrent brain metastasis after SRS/fSRT.

Comments11

As you mentioned, our conclusion was not easy to understand.

We adapted our conclusion.

Round 2

Reviewer 2 Report

The authors have addressed most of the comments. I would suggest minor spell check of the presented text